# Querying Similar Multi-Dimensional Time Series with a Spatial Database

**Zheren Liu [1], Chaogui Kang [2,3,]* and Xiaoyue Xing [4]**

1   School of Remote Sensing and Information Engineering, Wuhan University, 129 Luoyu Road,
    Wuhan 430079, China; zheren.liu@whu.edu.cn
2   National Engineering Research Center of Geographic Information System, China University of Geosciences,
    68 Jincheng Street, Wuhan 430078, China
3   Hubei Luojia Laboratory, 129 Luoyu Road, Wuhan 430079, China
4   Institute of Remote Sensing and Geographic Information Systems, School of Earth and Space Sciences,
    Peking University, 5 Yiheyuan Road, Beijing 100871, China; xyxing@pku.edu.cn
*   Correspondence: kangchaogui@cug.edu.cn; Tel.: +86-027-6788-3061

**Abstract:** Similar time series search is one of the most important time series mining tasks in our daily life. As recent advances in sensor technologies accumulate abundant multi-dimensional time series data associated with multivariate quantities, it becomes a privilege to adapt similar time series searches for large-scale and multi-dimensional time series data. However, traditional similar time series search methods are mainly designed for one-dimensional time series, while advanced methods applicable for multi-dimensional time series data are largely immature and, more importantly, are not friendly to users from the domain of geography. As an alternative, we propose a novel method to search similar multi-dimensional time series with spatial databases. Compared with traditional methods that often conduct the similarity search based on features of the raw time series data sequence, the proposed method stores multi-dimensional time series as spatial objects in a spatial database, and then searches similar time series based on their spatial features. To demonstrate the validity of the proposed method, we analyzed the correlation between temporal features of the raw time series and spatial features of their corresponding spatial objects theoretically and empirically. Results indicate that the proposed method can not only support similar multi-dimensional time series searches but also markedly improve its efficiency under many specific scenarios. We believe that such a new paradigm will shed further light on the similarity search in large-scale multi-dimensional time series data, and will lower the barrier for users familiar with spatial databases to conduct complex time series mining tasks.

**Keywords:** time series; multivariate; spatial database; similarity search

## 1. Introduction

Recent decades have seen a growing penetration of information technologies and sensory devices, which have brought forth an unprecedented spectrum of data that captures our ambient environments and societal dynamics [1,2]. Hidden in such data is a range of useful information to be distilled through data management and knowledge discovery methods. Time series mining provides a popular solution for such data accumulation and distillation that sit at the heart of many applications in finance [3], business [4], logistics [5], transportation [6], medicine [7], meteorology [8] and to name a few. Typical approaches include similarity search, clustering, classification, pattern discovery, and anomaly detection [9–12]. Among them, similarity search in time series data is fundamental to many advanced mining tasks [13].

Dating back to its early years, similarity search in time series data has heavily relied on either explicit features in a time sequence itself or implicit features derived from specific transformations on a time sequence [14]. However, recent years have seen a shift to

convert time series data into non-time series data for the sake of efficiently handling large-scale time series mining tasks [15–18]. Advances in the domain of image recognition and classification further pave a new way for converting time series data into image data to improve time series classification and imputation [19]. A few studies also attempted to translate time series data from/into spatial objects for feature extraction to facilitate similarity search [20,21].

The enhancement of the efficiency of similarity search is also a core task in time series mining. For one-dimensional time series data, search efficiency often depends on either length variability and/or size scalability [22]. It essentially requires algorithms for measuring the similarity of time series to be designed in a dynamically warping manner, and as a result, to be with expensive computational complexity [23]. Additionally, as data collection is prolonged, the increase of the length of the accumulated time series sequence would challenge the processes of both data storage and similarity calculation [24,25]. To increase computational speed, fast indexing is taken as a prerequisite of similarity search in large-scale time series data [26–28].

As recent advances in sensor technologies accumulated abundant multi-dimensional time series data associated with multivariate quantities [29], it becomes a privilege to adapt similar time series searches for large-scale and multi-dimensional time series data. Traditional solutions for addressing length variability and size scalability can be applied. However, it is nontrivial to handle such time series data because they often consist of many long sequences embedded in multiple dimensions (i.e., large in size, long in length, and high in dimensionality) [30] and embody complex correlations between the observed variables [31]. Moreover, advanced methods that are applicable for multi-dimensional time series data are still immature and, more importantly, are not friendly to users from the domain of geography.

Bearing these limitations in mind, we find that it is possible to address the problems of search efficiency and information loss facing similar multi-dimensional time series searches from a spatial perspective. First, a multi-dimensional space can embed and synthesize multiple variables by its very nature. Second, spatial objects can preserve characteristics of the raw time series as a whole without information loss. More importantly, the well-developed spatial database can provide various calculating and indexing functions that support effective similarity searches. As such, by proposing an alternative method that treats multi-dimensional time series data as spatial objects and implements the similarity search procedure using spatial indexes and analytic functions in the PostgreSQL object-relation database system with the PostGIS extension, we believe that such a new paradigm will shed further light on the similarity search in large-scale multi-dimensional time series data, and will lower the barrier for users familiar with spatial databases to conduct complex time series mining tasks.

## 2. Related Work

In the domain of similar time series search, there are generally two major strands of studies: one strand concerns accurate and elastic similarity measures between time series data and the other searches for efficient indexing methods to deal with large-scale and/or high-dimensional time series data. A brief review of these two topics is given in Sections 2.1 and 2.2.

### 2.1. Similarity Measures

Similar time series search often compares pair-wise time series based on predefined similarity measures to discover time sequences that share common characteristics either entirely (e.g., full sequence matching) or partially (e.g., subsequence matching) [32]. The former requires the best matches for a similarity query to be entire time sequences, and therefore is plausible for searching time series with equal lengths; the latter allows the best matches for a similarity query to be arbitrary subsequences of time series and therefore is flexible for searching short time series. Practically, similarity search often converges to

nearest neighbor search [33,34], which seeks to retrieve a given number of target sequences minimizing their dissimilarity to a query sequence in a time series database. In this study, we mainly focus on nearest neighbor search based on full sequence matching in time series data.

The most representative similarity measures for time series data mining include Euclidean distance, dynamic time warping (DTW), and longest common subsequence (LCSS), which have been widely integrated into toolkits such as the TSSEARCH [35] library. Mathematically, given two time sequences $A = ((a_1^1, \cdots, a_1^d), (a_2^1, \cdots, a_2^d), \cdots, (a_m^1, \cdots, a_m^d))$ and $B = ((b_1^1, \cdots, b_1^d), (b_2^1, \cdots, b_2^d), \cdots, (b_n^1, \cdots, b_n^d))$, their Euclidean distance, $L^{euc}(A, B)$, can be defined as

$$L^{euc}(A, B) = \sqrt{\sum_{j=1}^{d} \sum_{i=1}^{l} (a_i^j - b_i^j)^2} \tag{1}$$

in the case that $A$ and $B$ are of equal length $l$ (i.e., $l = m = n$), and where $d$ denotes the dimensionality of the two time sequences; $i$ and $j$ denote indices of the time slot and the observation dimension, respectively.

DTW and LCSS distances are often applied in the case that the given time series data are one-dimensional (i.e., $d = 1$). Fortunately, previous studies have already extended both to deal with multi-dimensional time series data [36]. Given two multi-dimensional time sequences $A$ and $B$, their DTW distance, $L^{dtw}(A, B)$, and LCSS distance, $L^{lcss}(A, B)$, can be defined as

$$\begin{aligned} L^{dtw}(A, B) = &L^{euc}(A_m, B_n) + \\ &\min(L^{dtw}(A', B), L^{dtw}(A, B'), L^{dtw}(A', B')) \end{aligned} \tag{2}$$

$$L^{lcss}(A, B) = \begin{cases} 0 & A \text{ or } B \text{ is empty} \\ 1 + L^{lcss}(A', B') & L^{euc}(A_m, B_n) < \epsilon \\ \max(L^{lcss}(A', B), L^{lcss}(A, B')) & others \end{cases} \tag{3}$$

where $A' = ((a_1^1, \cdots, a_1^d), (a_2^1, \cdots, a_2^d), \cdots, (a_{m-1}^1, \cdots, a_{m-1}^d))$; $B' = ((b_1^1, \cdots, b_1^d), (b_2^1, \cdots, b_2^d), \cdots, (b_{n-1}^1, \cdots, b_{n-1}^d))$; and $\epsilon$ is a constant threshold that defines a successful match between two data observations.

The advantage of Euclidean distance lies in its simplicity in both definition and calculation. However, the use of Euclidean distance is largely limited to time series data with equal length. It therefore often suffers to cope with data issues concerning noise, distortion, offset translation, and amplitude scaling that are inherent in time series [37]. In comparison, DTW and LCSS distances can measure the similarity between time series with such limitations. They dynamically offset time series to find their best matching counterpart, which does not essentially require the two time series to be of the same length. However, cases that result in large warping are not rare when applying DTW to offset noisy time series data. Therefore, it is often a requirement to introduce certain bandwidth constraints (such as the Sakoe–Chiba band and the Itakura parallelogram) into DTW to avoid large time warping [38,39]. Such bandwidth constraints are inherently considered in LCSS, which defines a time window within which the match between a given point from one time series to a point in another time series is allowed [40]. Considering that DTW distance often yields better performance than Euclidean distance in terms of both accuracy and robustness, we take it as the reference method in this study.

It is also noteworthy that the similarity of time series can be quantified in two different ways [41]. The strict similarity requires that the absolute values of variables in different time series are approximate at each sampling time point. However, non-strict similarity

requires that only the trends of time series are similar. Therefore, trend similarity is often calculated on the normalized time series $\tilde{A} = ((\tilde{a}_1^1, \cdots, \tilde{a}_1^d), \cdots, (\tilde{a}_m^1, \cdots, \tilde{a}_m^d))$, where

$$\tilde{a}_i^j = \frac{a_i^j - \bar{a}^j}{std(a^j)} \tag{4}$$

Specifically, for full sequence matching, $\bar{a}^j$ and $std(a^j)$ are the temporal average and the standard deviation of the $j$th dimension of the whole time series; For subsequence matching, $\bar{a}^j$ and $std(a^j)$ are the average and the standard deviation of the sequence segment within a given sliding window. A lot of studies on time series mining usually normalize time series to highlight the trend similarity and neglect the difference of magnitude levels among the time ranges [42]. Nonetheless, it is worth noting that the strict similarity search is necessary under situations where the absolute values of data observations are also meaningful, such as traffic capacity analysis and flow pattern mining [43].

### 2.2. Indexing Methods

Although DTW and LCSS distances are plausible measures for the similarity between time series in practice, they often suffer from high computational complexity when dealing with massive time series data. Fortunately, by indexing time series, we can accelerate the similarity search by avoiding the full table scan and reducing the number of times of pairwise time series comparisons. Indeed, using time series indexing to filter out significantly dissimilar sequences has become the preliminary step for further matching time series accurately.

#### 2.2.1. Indexing One-Dimensional Time Series

To speed up the similarity search for one-dimensional time series, many indexing methods have been developed. Among them, generic multimedia indexing (GEMINI)- and lower-bounding-based methods are two popular methods that can obtain accurate result sets with no false dismissal.

In detail, the GEMINI framework mainly includes the following steps [44]:

(1) Extract time series features through feature transformation and map a sequence to a point in the feature space;

(2) Build spatial indexes to organize the spatial points in feature space;

(3) When a query request is received, search close points to the query point in the feature space using the spatial index to obtain the candidate set;

(4) Calculate the similarity between the query sequence and the candidate sequences to obtain the result set.

Following the GEMINI framework, different feature transformation and spatial indexing methods have been developed. For example, previous studies performed discrete Fourier transform (DFT) [45], Haar wavelet transform [46], discrete Cosine transform (DCT) [47], piece-wise aggregate approximation (PAA) [26] and Chebyshev polynomials (CHEB) [48] on time series, and built R*-tree indexes [49] to organize the feature points. Despite that R*-tree is an efficient method for organizing feature points in low-dimensional space, its pruning ability degenerates when the number of feature dimensions increases [50]. To address this limitation, improved indexing methods, such as TV-tree [51], X-tree [52], dynamic splitting tree (DSTree) [53] and adaptive data series index (ADS Index) [54], have been proposed to organize feature points in high-dimensional space to achieve efficient retrieval.

Another efficient way to index the feature points is using the lower-bounding technique. According to the lower-bounding distance lemma [55], there would be no false dismissals if the following formula is satisfied:

$$L^{feature}(A, B) \leq L^{data}(A, B) \tag{5}$$

where $L^{feature}(A, B)$ and $L^{data}(A, B)$ are distances between two time series in the feature space and the data space, respectively. As such, proper lower-bounding distance functions need to be designed to satisfy the lemma to achieve an exact search. A typical lower-bounding distance between time series is the cumulative distance of their envelope bounds built by sliding windows. Commonly used envelope-based lower bounds include *LB_Yi* [56], *LB_Kim* [57], *LB_Keogh* [27], *LB_Rotation* [58], etc. Among them, *LB_Keogh* is mostly used due to its easy calculation and good tightness. More details for the lower-bounding techniques can be found in ref. [37].

Recently, the growing data volumes bring new challenges to the computational efficiency of similar time series searches. Consequently, approximate search algorithms are receiving growing attention due to their high efficiency while the loss of accuracy can be controlled by predefined function parameters [59]. In particular, locality-sensitive hashing (LSH) and its variants are the most widely used methods for dealing with large-scale and high-dimensional data indexing problems, including the weighted locality-sensitive hashing (WLSH) technique for streaming time series data [60], LSH-based time sub-series retrieval [61], and LSH-based similar strings search by converting time series to character strings [17]. In general, those methods use a family of hashing functions to map time series into buckets, and similar time series are mapped in one bucket with higher probability than dissimilar ones. Only related buckets will be visited in a query, therefore candidate set contains a limited number of series, and the search cost is reduced [62].

### 2.2.2. Indexing Multi-Dimensional Time Series

Indexing methods for one-dimensional time series are difficult to directly apply to index multi-dimensional time series. For GEMINI-based methods, they are unable to do the feature transformation on multi-dimensional time series [25]. Various solutions have been proposed, including transforming multi-dimensional series to one-dimensional series through dimensionality reduction methods such as principal component analysis (PCA) before building index [63,64], evaluating the similarity of the multi-dimensional time series in each dimension and using the Borda voting method to achieve similarity searching [65], and summarizing multi-dimensional features at each time point to one representative value as the center sequence for establishing specific lower bounds such as *LB_CMK* [31]. However, all these methods need to transform the original multi-dimensional time series in advance. As a result, they often suffer from substantial information loss and high cost of transformation time.

Besides one-dimensional time series, lower-bounding constraint methods can also be applied to index multi-dimensional time series data with certain extensions. For example, a multi-dimensional time series search can be sped up by dividing the minimum bounding envelope (MBE) into minimum bounding rectangles (MBR) and then building an R-tree to index the resulting MBRs [66]. However, as the number of dimensions increases, these indexing methods will be affected by the curse of dimensionality and might suffer significant performance degradation potentially to a degree even worse than sequential scanning [67]. More importantly, building MBR for multiple subsequences and calculating MBR distances require extra storage overhead and result in a high time cost of computation, especially when the operations are not facilitated by well-implemented spatial functions and mature platforms for massive spatial relation calculation.

Approximate search algorithms such as LSH have been proven to be efficient for fast one-dimensional time series search, but the use of these methods on multi-dimensional data is still an emerging and hot topic [68]. For example, a previous study designed an LSH-based index to accommodate multi-dimensional time series retrieval [69]. However, those approximate methods largely remain immature. Hash functions applied for the multi-dimensional vector are different at each time point, so the time series are converted to a sequence of independent hashing values instead of a sequence with temporally correlated variables as a whole. Therefore, it is necessary to design user-friendly methods to achieve fast similarity searches for multi-dimensional time series.

To fill the gap, this study aims to develop a novel time series similarity search method that will be easy-to-use for researchers from the domain of geography (who are familiar with spatial databases). The main idea is to represent multi-dimensional time series as spatial objects and use built-in functions and indexes of spatial databases to quantify their similarity and achieve efficient search from a spatial perspective. We will analyze the relationship between time series and spatial objects to verify whether the proposed transformation can inherit critical properties of the raw time series in spatial characteristics of its spatial counterparts. Moreover, we implement the proposed method on the PostGIS-extended PostgreSQL database and empirically compare the ability and efficiency of the proposed method with several existing similarity search methods under various scenarios.

## 3. Methodology

The proposed spatial database-based time series similarity search method first converts the original multi-dimensional time series into spatial objects in feature spaces and establishes the associations between the temporal and spatial features. Then, spatial queries and calculations supported by spatial databases can satisfy the requirements for effective similarity search on multi-dimensional time series. Details on the associations and the searching approaches (for 2D time series data) will be introduced in Sections 3.1 and 3.2.

### 3.1. Association between Time Series and Spatial Objects

A valid conversion from time series to spatial objects should provide an explicit corresponding relation between temporal characteristics of the raw multi-dimensional time series and spatial characteristics of its corresponding spatial object. It will ensure that the similarity search of the original time series based on features of their spatial counterparts could be effective. The multi-faceted corresponding relation between temporal features of multi-dimensional time series and spatial features of database objects is clarified below.

#### 3.1.1. Conversion from Time Series to Spatial Objects

The data structure of multi-dimensional time series is highly identical to that of spatial trajectories of a moving object in multi-dimensional spaces. Hereafter, we will elaborate on the proposed method based on the two-dimensional time series for the sake of simplicity and conciseness.

In general, the temporal changes of two variables $x$ and $y$ can be organized as a multi-dimensional time sequence in the form of $((x_1, y_1), (x_2, y_2), \ldots, (x_m, y_m))$. If an explicit time dimension $t$ is further added to record the timestamps of each data observation, the time sequence can be reformed as $((x_1, y_1, t_1), (x_2, y_2, t_2), \ldots, (x_m, y_m, t_m))$. Indeed, such a form has also been widely used to represent the trajectory of moving objects in a two-dimensional space $\mathbb{R}^{(x,y)}$. It implies that the temporal changes of two variables $x$ and $y$ can be intuitively captured by the spatial trajectories in a two-dimensional feature space $\mathbb{R}^{(x,y)}$ within which the two feature dimensions designate the observed variables $x$ and $y$.

As illustrated in Figure 1, the values of the two variables $x$ and $y$ at each time point $t$ can be taken as the coordinates of a corresponding point in the feature space $\mathbb{R}^{(x,y)}$. This conversion ensures that the number of observed variables in the time series is consistent with the dimension of the feature space. Moreover, since the temporal information is inherited in the ordering of the points in each spatial object, time series collected in different time periods also become comparable in spatial database-based time series similarity search after the conversion.

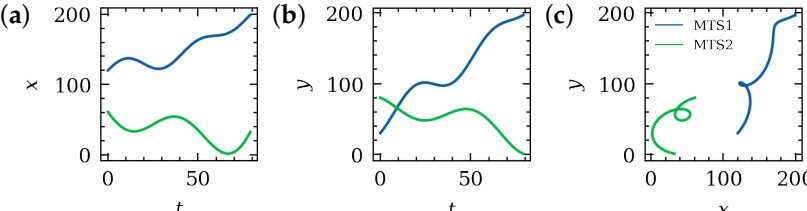

**Figure 1.** Transformation from multi-dimensional time series (MTS) to spatial objects. (**a**) Time series in dimension $x$; (**b**) Time series in dimension $y$; (**c**) The corresponding spatial objects embedded in the two-dimensional space $\mathbb{R}^{(x,y)}$.

### 3.1.2. Temporal Features and Its Spatial Counterparts

As aforementioned, it is essential to understand the association between features of time series and that of their spatial counterparts to discover representative spatial features that can support similar time series searches in spatial databases. In this study, we focus on the centroid, the MBR, and the shape difference between spatial objects.

The centroid $(\bar{x}, \bar{y})$ of a spatial object mathematically coincides with the temporally averaged magnitudes $\sum_{i=1}^{m} x_i / m$ and $\sum_{i=1}^{m} y_i / m$ of the observed variables. It suggests that we can measure the difference in the average levels of all the observed variables between time series using the Euclidean distance between the centroids of their spatial counterparts (Figure 2).

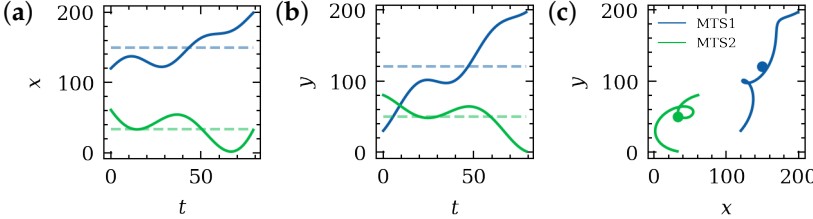

**Figure 2.** Similarity between two-dimensional time series by the distance between centroids of the corresponding spatial objects in the two-dimensional space. (**a**) Mean value of time series in dimension $x$; (**b**) Mean value of time series in dimension $y$; (**c**) Euclidean distance between the centroids of the corresponding spatial objects.

In spatial databases, the MBR has been intensively used for the optimization of spatial queries. It is often generated and applied to build the spatial index for each spatial object. As shown in Figure 3c, the extension (i.e., width and height) of the spatial objects in space $\mathbb{R}^{(x,y)}$ coincide with the ranges of temporal variations for variables $x$ in Figure 3a and $y$ in Figure 3b, respectively. The MBR characterizes the upper and lower bounds in each feature dimension, so that the constraints on the maximum and minimum values of the time series can be transformed into the constraints on spatial extensions of the MBR.

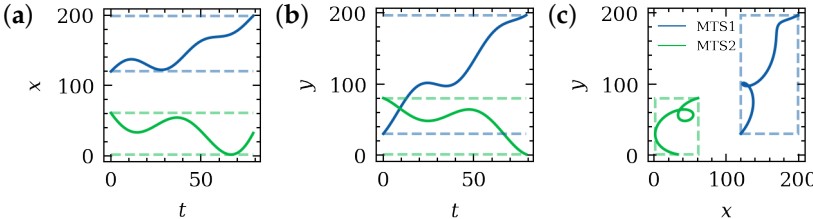

**Figure 3.** Similarity between two-dimensional time series by the MBR of the corresponding spatial objects in the two-dimensional space. (**a**) Minimum/maximum value of time series in dimension $x$; (**b**) Minimum/maximum value of time series in dimension $y$; (**c**) MBRs of the corresponding spatial objects.

The shape difference between spatial objects can be taken as a proxy for the dissimilarity between the raw time series. Various measures have been developed to quantify the shape difference between spatial objects, such as Euclidean distance, edit distance, Hausdorff distance, and one-way distance [70–72]. In this study, a proper distance measure should measure the distances between polygonal curves of spatial objects dynamically and accurately, and, more importantly, can be directly calculated using SQL statements. We select Fréchet distance as the indicator of shape difference in that it is a built-in function in many spatial databases while DTW distance is not. It dynamically matches points on two curves in a warping way, and then uses the largest distance of all the matched point pairs as the distance measure.

Given two spatial objects $A = ((a_1^1, a_1^2, \cdots, a_1^d), (a_2^1, a_2^2, \cdots, a_2^d), \cdots, (a_m^1, a_m^2, \cdots, a_m^d))$ and $B = ((b_1^1, b_1^2, \cdots, b_1^d), (b_2^1, b_2^2, \cdots, b_2^d), \cdots, (b_n^1, b_n^2, \cdots, b_n^d))$, which are in the same form as the multi-dimensional time series defined in Section 2.1, their Fréchet distance, $L^{fréch}(A, B)$, can be defined as

$$
\begin{aligned}
L^{fréch}(A, B) = \max(&L^{euc}(A_m, B_n), \\
&\min(L^{fréch}(A, B'), L^{fréch}(A', B), L^{fréch}(A', B')))
\end{aligned}
\tag{6}
$$

Comparing Equations (2) and (6), we notice that the recursive calculation formula of the Fréchet distance and that of the DTW distance are quite similar. In terms of computational complexity, the DTW distance and the Fréchet distance are the same as each other, which is $O(mn)$. Their difference merely lies in whether the final results are derived from the summation or the maximum distance of all the matched point pairs. Therefore, it is plausible to take the Fréchet distance between spatial objects as a proxy for the DTW distance between multi-dimensional time series.

### 3.2. Similar Time Series Search in Spatial Database

Based on the associations between time series and spatial objects, we search similar spatial objects in spatial databases to find similar time series. Compared with time series search, spatial search has several advantages. First, it is easier to build a spatial index than a time index in that a spatial index only relies on the original data while feature transformation and boundary calculation are required for time indexing. Second, spatial databases can store each multi-dimensional time series as a single spatial object, so they have the great capability of storing time series data in an object-oriented manner. Last but not the least, a variety of spatial operation functions has been integrated into spatial databases which enable us to efficiently calculate spatial features closely related to temporal characteristics of the raw time series.

As aforementioned in Section 2.1, we will describe the similar time series search process in terms of strict similarity and trend similarity separately in the following sections.

#### 3.2.1. Top-*k* Search for Strictly Similar Time Series

The strict similarity search filters time series based on different temporal features step by step. First, according to the definition of strict similarity, only the time series associated with similar magnitude can be similar. Thus, the average level of the time series is analyzed in the first step (Figure 4b). For time series at the same average level, temporal variations of the time series might be distinctly different. It indicates that the range of temporal variations should be considered secondly (Figure 4c). The candidate set obtained from the above two steps will exclude the time series largely deviating from the query sequence in terms of the average value and range of variations. To refine the final top-*k* similar time series, we can calculate the shape difference between the candidate set and the query sequence in the last step (Figure 4d).

Specifically, we can conduct the above filtering process sequentially based on the centroid, the MBR, and the Fréchet distance of spatial objects in spatial databases. It is intuitive to implement filters of the average levels, the ranges of variation, and the

differences of the curve shape in spatial databases according to the association between temporal and spatial features.

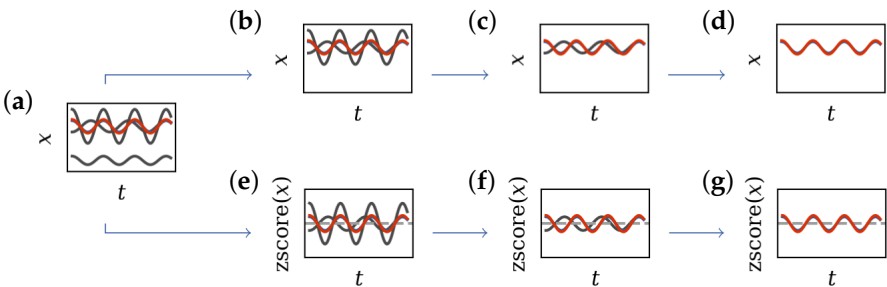

**Figure 4.** Procedures of the proposed similar time series search method. (**a**–**d**) Strictly similar time series search; (**a**,**e**,**f**,**g**) Search of time series with similar trend.

Given a time series dataset $\mathbb{S}$ stored in the spatial database and a query sequence $Q$, the procedure of searching the top-$k$ strictly similar time series to $Q$ is as follows:

(1) Calculate the difference between the average value of each time series in $\mathbb{S}$ and that of the query sequence $Q$ using built-in spatial functions *ST_Centroid* and *ST_Distance* in the spatial database. The top-$5k$ nearest time series are retrieved as a candidate set $\mathbb{S}_{5k}$. To speed up the top-$5k$ search, the centroid of each spatial object is spatially indexed.

(2) Calculate the areal difference between the range of temporal variation in each time series in $\mathbb{S}_{5k}$ and that of $Q$ using the built-in MBR functions *ST_Envelope* and *ST_Area*. The top-$2k$ nearest time series are retrieved from the candidate set $\mathbb{S}_{5k}$ to compose a refined candidate set $\mathbb{S}_{2k}$.

(3) Calculate the accurate similarity between time series in $\mathbb{S}_{2k}$ and $Q$. Recall that PostGIS does not provide the DTW distance function, we use the built-in Fréchet distance function *ST_FrechetDistance* as an alternative to the DTW distance as mentioned in Section 3.1.2. After sorting the time series in $\mathbb{S}_{2k}$ according to their Fréchet distance to $Q$, the top-$k$ nearest objects are finally obtained as the result set $\mathbb{S}_k$.

### 3.2.2. Top-$k$ Search for Time Series with Similar Trend

The trend similarity search is useful for finding time series that share primarily the same trend, while their ranges of temporal variation are not at the same level. Under this circumstance, the original series need to be normalized in advance. The procedure of trend similarity search is basically the same as that of the strict similarity search, except for the first step. Since the temporal averages of all the variables in the time series are zeros after $z$-score normalization (Figure 4e), the centroids of all spatial objects are located at the origin of coordinates in the multi-dimensional feature space. Therefore, only step 2 (Figure 4f) and step 3 (Figure 4g) introduced in Section 3.2.1 are considered in the search of time series with a similar trend. The procedure of searching the top-$k$ time series with a similar trend to a given query sequence is as follows:

(1) Translate and rescale the query sequence $Q$ and time series in $\mathbb{S}$ using built-in transformation operations *ST_Translate* and *ST_Scale* in the spatial database. Such transformations are equivalent to the $z$-score normalization of the raw time series. The normalized time series dataset $\tilde{\mathbb{S}}$ is obtained. Please note that this step is for data pre-processing and its time cost should be compared with the pre-processing of the raw time series data.

(2) Filter $\tilde{\mathbb{S}}$ in the same way as the second step of strict similarity search, and obtain the candidate set $\mathbb{S}_{2k}$.

(3) Filter $\mathbb{S}_{2k}$ in the same way as the final step of strict similarity search, and obtain the result set $\mathbb{S}_k$.

For either strict similarity or trend similarity, the major advantage of converting time series to spatial objects lies in that we can effectively use spatial indexes and well-implemented spatial operations in spatial databases to support fast similarity searches.

In contrast, if these measures of difference are calculated directly on the raw time series, we need to filter and sort candidate time series based on sequential scan and comparison, which cannot meet the requirements of fast querying in large-scale datasets.

## 4. Experiments and Evaluations

In this study, we implemented the proposed similarity search method in the open-source database PostgreSQL 11.5 and PostGIS 2.5. All experiments were conducted on a PC machine with a 2.90 GHz Intel(R) Core(TM) i5-10400 CPU and 16 GB RAM. Considering that DTW distance is the most popular measure of shape difference between time series, the DTW-based similarity search method introduced by Rakthanmanon et al. [73] is taken as the reference method for comparative analyses, and the *LB_Keogh* lower bound [20] is used to speed up the search process. The original program of the DTW-based method only supports a top-1 similarity search for one-dimensional time series, and we thus re-implement it in the C language to support a top-*k* similarity search for multi-dimensional time series. All codes are available at https://figshare.com/s/f2f74937ed2a4f6a5ecf (accessed on 22 October 2022). Please note that, since the GEMINI-based similarity search method is not suitable for multi-dimensional time series, it is not considered to be a reference method in our experiments.

### 4.1. Performance of the Proposed Method

We evaluate the proposed similarity search method on several real-world datasets. Specifically, four publicly available datasets ("CharacterTrajectories", "EthanolConcentration", "Libras", and "PenDigits") from the UCR Time Series Classification Archive [74], and a privately collected human mobility dataset in Beijing ("BeijingDynamics") are thoroughly evaluated. Descriptive information of the five datasets is listed in Table 1.

**Table 1.** Descriptive information of the datasets for evaluation.

| Name | $\|\mathbb{S}\|$ | $l$ | $d$ | Description |
| --- | --- | --- | --- | --- |
| CharacterTrajectories | 1422 | 182 | 3 | Handwriting character trajectories captured using a WACOM tablet. Each instance is a 3-dimensional pen tip velocity trajectory. |
| EthanolConcentration | 261 | 1751 | 3 | Raw spectra taken of water-and-ethanol solutions with different concentrations. The wavelength range of the recorded spectra is from 226 nm to 1101.5 nm, with a sampling interval of 0.5 nm. |
| Libras | 180 | 45 | 2 | Hand movement represented as a bi-dimensional curve performed by the hand in a period of time. The curves were obtained from videos of hand movements. |
| PenDigits | 7494 | 8 | 2 | Handwritten digit instances made up of the $x$ and $y$ coordinates of the pen traced across a digital screen. |
| BeijingDynamics | 1107 | 1116 | 2 | Hourly variation of in/outflow of each traffic analysis zone in Beijing. The data are derived from signaling records between 1 July 2018 to 31 August 2018 provided by China Unicom. |

We randomly select 10% samples from each dataset as the query sequence, and then conduct the top-1%, 5%, and 10% similarity search for these samples. After the result sets are obtained, we quantify the consensus, $\rho$, between the result of the proposed method and that of the reference method as

$$\rho = \frac{\|\mathbb{S}_k \cap \mathbb{S}'_k\|}{k} \qquad (7)$$

where $\mathbb{S}_k$ is the result set obtained by the proposed search method; $\mathbb{S}'_k$ is the result set obtained by the DTW-based search method; $\| * \|$ is the number of elements in a set; and $k$ is the retrieval number. Additionally, we quantify the efficiency improvement (or the speedup ratio), $\eta$, of the proposed method over the reference method as

$$\eta = \frac{T'}{T} \qquad (8)$$

where $T$ and $T'$ are the computational time of the proposed method and that of the reference method, respectively.

As listed in Table 2, the performance of the proposed search method varies in datasets with different numbers of time series and different lengths of time observations. The consensus of the results of the strict similarity search is over 40% in most cases, except for the top-1% similarity search in "CharacterTrajectories" and "PenDigit". These two datasets and "BeijingDynamics" have large amounts of time series, and their consensus levels mainly increase with the retrieval number $k$. Whereas, for small datasets such as "EthanolConcentration" and "Libras", the top-1% similarity search achieves high consensus. Results for trend similarity show the same regulations, but the overall consensus level is lower. For efficiency improvement, the proposed method can speed up the search process in datasets of short time sequences, such as "Libras" and "PenDigit". However, the speedup ratio decreases along with the size of the final result set. The opposite changing patterns of consensus and speedup ratio with retrieval number suggest that a trade-off between retrieval accuracy and computational efficiency is considered in the proposed method.

**Table 2.** Comparisons between the proposed method and the reference method.

| Name | Indicator | Strict Similarity | | | Trend Similarity | | |
|---|---|---|---|---|---|---|---|
| | | 1% | 5% | 10% | 1% | 5% | 10% |
| CharacterTrajectories | $\rho$ | 0.384 | 0.458 | 0.452 | 0.113 | 0.180 | 0.229 |
| | $\eta$ | 1.844 | 0.434 | 0.255 | 1.701 | 0.443 | 0.255 |
| EthanolConcentration | $\rho$ | 0.654 | 0.598 | 0.663 | 0.500 | 0.192 | 0.188 |
| | $\eta$ | 0.349 | 0.140 | 0.106 | 0.476 | 0.158 | 0.115 |
| Libras | $\rho$ | 1.000 | 0.463 | 0.454 | 1.000 | 0.222 | 0.228 |
| | $\eta$ | 8.887 | 1.509 | 0.776 | 7.944 | 1.499 | 0.790 |
| PenDigits | $\rho$ | 0.197 | 0.380 | 0.479 | 0.057 | 0.161 | 0.259 |
| | $\eta$ | 8.958 | 2.474 | 1.358 | 2.933 | 1.778 | 1.200 |
| BeijingDynamics | $\rho$ | 0.435 | 0.608 | 0.688 | 0.114 | 0.133 | 0.192 |
| | $\eta$ | 0.358 | 0.152 | 0.110 | 0.711 | 0.211 | 0.165 |

Recall that, in the "BeijingDynamics" dataset, the number of time series is large and the length of each time series is long. Considering that each time series in this dataset is explicitly associated with a specific spatial location, we further explore the spatial and temporal patterns of the time series in the final result set in the next section.

*4.2. In-Depth Evaluation on "Beijingdynamics" Dataset*

In detail, the "BeijingDynamics" dataset consists of time series data associated with the TAZs within the 6th Ring Road of Beijing, China. Excluding TAZs within which human mobility flux is not observed, the final dataset contains a total of 1107 time series. All the time series are observed hourly for 62 consecutive days. Since only a few trips were observed between 0:00 a.m. and 6:00 a.m. of each day, we exclude this period and combine the rest data in 62 days. The length of the combined time series is 1116. Each time series consists of two observation dimensions that reflect human mobility patterns from the perspective of inflow and outflow, respectively. The spatial distribution of the daily averaged human mobility flux in the case study area is shown in Figure 5.

To evaluate the relationship between the consensus, $\rho$, the computational efficiency, $\eta$, and the size of the result set, $k$, we conduct the top-1, 5, 10, 20, 50, and 100 similarity search for 200 randomly sampled time series from the dataset using the proposed method and the reference method, respectively.

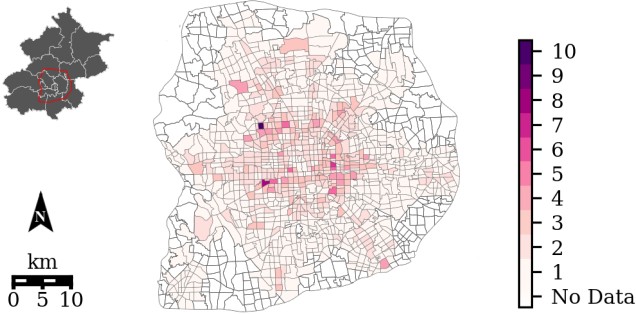

**Figure 5.** Overview of the case study area in Beijing, China. The color in the Choropleth represents the number of human mobility flux captured by the mobile phone signaling data. Each unit denotes 10,000 people. Each polygon covers a traffic analysis zone.

### 4.2.1. Comparison from a Stepwise Perspective

As proposed, the process of similar time series search can be split into two separate steps: (1) Use spatial features and/or lower bounds as constraints for pruning in advance, which involves all calculation and filtering steps before measuring the exact distance for the candidate set, $\mathbb{S}_{2k}$; (2) Calculate exact distance for time series in the candidate set, $\mathbb{S}_{2k}$, using Fréchet and/or DTW distance algorithms to obtain the result set, $\mathbb{S}_k$. As such, we further inter-exchange the spatial pruning and the lower-bounding constraint in the first step as well as the Fréchet distance and the *LB_Keogh* constrained DTW distance in the second step of the similarity search process to evaluate the consensus and efficiency improvement between the proposed method and the reference method more fairly and comprehensively.

As shown in Figure 6, despite the different choices of similarity measures in the second step, the consensus levels $\rho$ show the same relationship with the retrieval number $k$. Specifically, the overall consensus increases when the distance measures in the proposed method and the reference methods are identical. Moreover, the consensus levels increase from top-5 to top-100 similarity search under all circumstances. Although the result set obtained from the proposed method and that from the reference method are not the same, their consensus can exceed 0.5 in certain scenarios, such as the top-1 similarity search for both strict similarity search and trend similarity search, as well as the top-50 and 100 searches for strict similarity. In the scenario that the distance measure in the proposed method is the same as that in the reference method, their consensus values are enhanced. In particular, consensus between the result set from the proposed method and the result set from the traditional Fréchet method remains the highest for strict similarity, while consensus between the spatial-pruned DTW method and the reference method shows the highest value in the search of time series with a similar trend.

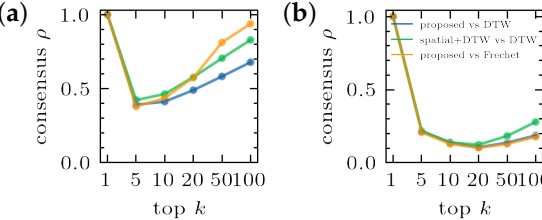

**Figure 6.** Consensus between the proposed method, the reference method, and their variants. (**a**) Strictly similar time series search; (**b**) Search time series with similar trend.

The efficiency improvements of the proposed method also change along with the retrieval number $k$. As shown in Figure 7a,d, if the retrieval number is small ($k < 5$), the proposed method is superior to the reference method in its computational efficiency. However, as the retrieval number $k$ increases, the efficiency of the proposed method decreases remarkably and becomes inferior to the reference method. This result can be largely attributed to the fact that the size of the candidate set $\mathbb{S}_{2k}$ in the proposed method grows proportionally with $k$ and the computationally expensive Fréchet distance in the second step need to be calculated for these candidates. It is noteworthy that the computational complexity of the Fréchet distance and that of the DTW distance are theoretically the same. However, in this study *LB_Keogh* is used as a constraint for accelerating the C program implemented DTW-based search method, while the Fréchet distance function in PostgreSQL is not optimized (Figure 7b,e). On the contrary, in the first step of pruning, we find that spatial pruning can effectively replace lower-bounding constraints in the original time series similarity search. Considering that both the MBR in two-dimensional space and the *LB_Yi* lower bound of one-dimensional series share the same idea of max-min boundary for constraint, we also compare the two methods in terms of the pruning efficiency. Results in Figure 7c,f indicate that the MBR-based spatial features are more effective than the *LB_Yi* lower bound in the pruning step.

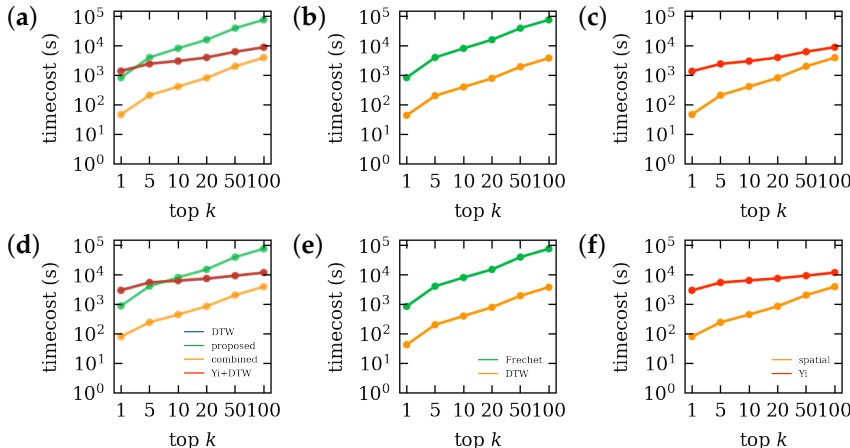

**Figure 7.** Computational cost of similarity search on time series in different steps. (**a**–**c**) Strictly similar time series search; (**d**–**f**) Search time series with the similar trend; (**a**,**d**) The overall computational time; (**b**,**e**) The computational time of the distance calculation step; (**c**,**f**) The computational time of the pruning step.

In Figure 7, the two additional methods that ex-changeably combine advantageous steps of the proposed method and the reference method further reveal the ability and potential of the proposed method. The combined method that uses spatial features for pruning and generating set $\mathbb{S}_{2k}$ and calculates exact *LB_Keogh* constrained DTW distance for obtaining the result set $\mathbb{S}_k$ always yields lower time cost than other methods. In other words, the computational efficiency can be significantly improved if spatial features are used for pruning before the expensive distance calculation. However, the time cost of similarity search that combines *LB_Yi* and *LB_Keogh* pruning (Yi+DTW) is quite similar to the reference method that uses only *LB_Keogh* pruning but is still significantly higher than the combined method. This fact again suggests that the pruning power of *LB_Yi* is weak and the spatial features in the database play a more important role in the process of search acceleration. Intuitively, in case the retrieval number, *k*, is large, the pruning will play a less important role in the first step because the candidate set $\mathbb{S}_{2k}$ might contain most of the time series in the raw set $\mathbb{S}$. Consequently, the time cost of the combined method will gradually approach that of the reference method as the retrieval number increases.

### 4.2.2. Comparison from a Spatial Perspective

The consensus indicator distinguishes if the result set of the proposed method and that of the reference method are the same or not. Under closer scrutiny, we further visualize the spatial and temporal patterns of each time series similar to the query sequence, and evaluate the spatio-temporal similarity between the two result sets with the assistance of visual analytics.

Although the retrieved time series in $\mathbb{S}_k$ cannot exactly match those in $\mathbb{S}'_k$ due to the slight difference between DTW distance and Fréchet distance, we find that they are highly similar in terms of both of their curve shapes and spatial distributions. Taking Temple of Heaven (Figure 8a) and Sanlitun (Figure 8d) as examples, the top-20 similar time series for the two TAZs are extracted and visualized in Figure 8b,c and Figure 8e,f, respectively. Only the curves of outflow variation are drawn for illustration purposes. The spatial distributions of the corresponding TAZs are also highlighted in the maps (Figure 8a,d). Since the Temple of Heaven is a famous scenic spot and Sanlitun is one of the greatest business centers in Beijing, their functions and human mobility patterns are different. For result sets obtained by both search methods, the variation pattern of the resulting time series is very similar to the original query sequence. It suggests that although the proposed method cannot extract completely consistent results with the reference method, it is still able to correctly find highly similar time series to the query sequence. In addition, the spatial distribution and street-views of TAZs in the result sets also confirm the capability of the proposed method. As listed in Table 3, the physical characteristics of the Top-20 TAZs that are similar to Temple of Heaven and Sanlitun determined by the DTW method and our proposed method are visually similar to each other, while their similarities to randomly selected TAZs are much lower. Please note that we collect the street-view images of every 200 m × 200 m grid within a TAZ, and select the scene with the most frequent appearance to represent its typical physical characteristics. The spatio-temporal difference between the result set $\mathbb{S}'_k$ obtained from the reference method and the result set $\mathbb{S}_k$ obtained from the proposed method for each specific TAZ provides additional evidence for the fair comparison between the two methods. Therefore, we further explored the validity of the proposed method from the perspective of spatial and temporal distance deviations.

Specifically, we first define an indicator named boundary distance, which is the maximum DTW distance between the query sequence and the time series in the result sets obtained by the reference method. Then, for each time series in the result sets obtained by the proposed method, its DTW distance to the query sequence is calculated to check if this DTW distance would deviate from the boundary distance. We consider a similar time series as a good hit if its DTW distance does not exceed the boundary distance of more than 20%. For the top-5, 10, 20, 50, and 100 similar time series searches, the proportions of good hits are 0.742, 0.792, 0.854, 0.886, and 0.913, respectively. It indicates that the deviation

between the results of the proposed search and that of the reference method is within an acceptable range for this dataset.

To quantify the spatial deviation of two result sets, we define an indicator named average offset distance between sets. For each TAZ in $\mathbb{S}_k$, its distance to the nearest TAZ in $\mathbb{S}'_k$ is calculated, and then the mean value of the distances for all TAZs is finally taken as the average offset distance between the two sets. Results for cases with different retrieval numbers, $k$, and different search strategies (i.e., strict similarity and trend similarity) are shown in Figure 9. Comparing the offset distances of result sets with that of $k$ randomly selected TAZs, the result set $\mathbb{S}_k$ shows significantly different distributions of offset distances with shorter and more concentrated offset values. Given that the average distance between adjacent TAZs in the case study area is 1.35 km, the average offset distance of the proposed method is about 1 to 4 TAZs, which is acceptable. Moreover, as the retrieval number $k$ increases, the spatial difference of the result sets obtained by the two methods gradually dismisses. For the case of searching top-100 similar series of a query sequence, the average offset distance of most of the TAZs can be reduced to less than 2 km, which is about 2 TAZs apart. In this sense, it seems reliable to search similar time series using spatial databases.

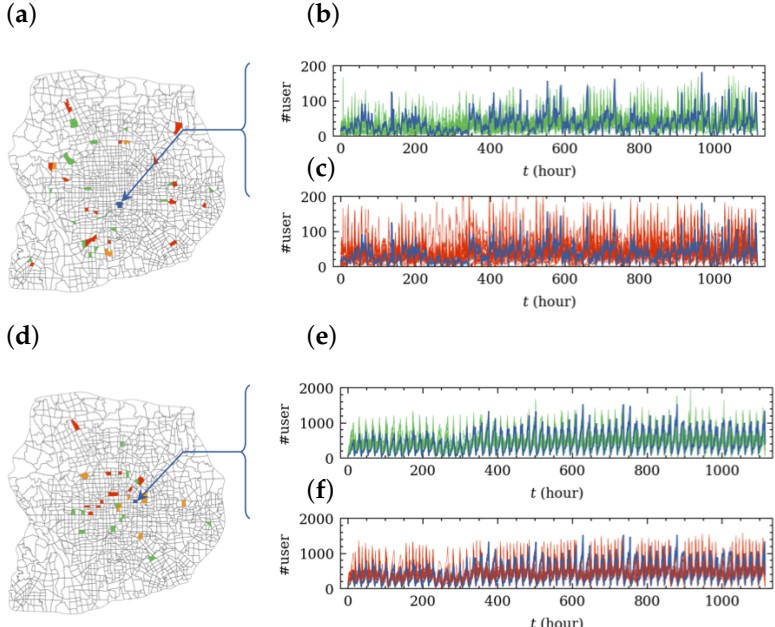

**Figure 8.** Consensus between the spatial and temporal patterns of TAZs retrieved by the proposed method and the reference method. (**a**) The spatial distributions of the retrieved TAZs similar to Temple of Heaven; (**b**) Temporal patterns of the similar time series retrieved from the reference method; (**c**) Temporal patterns of the similar time series retrieved from the proposed method; (**d**) The spatial distributions of the retrieved TAZs similar to Sanlitun; (**e**) Temporal patterns of the similar time series retrieved from the reference method; (**f**) Temporal patterns of the similar time series retrieved from the proposed method.

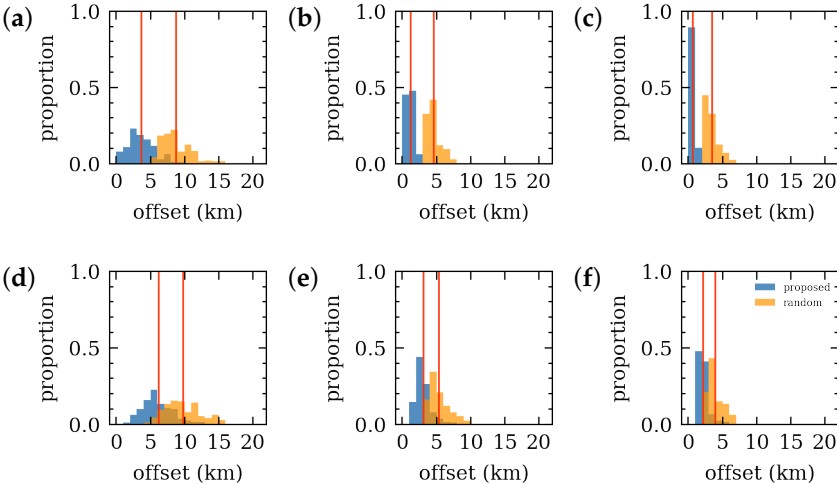

**Figure 9.** Offset distance between the spatial distributions of TAZs retrieved from the proposed method and the reference method. (**a**) Distribution of the offset distance between top-10 strictly similar time series from the two methods; (**b**) Distribution of the offset distance between top-50 strictly similar time series from the two methods; (**c**) Distribution of the offset distance between top-100 strictly similar time series from the two methods; (**d**) Distribution of the offset distance between top-10 time series with a similar trend from the two methods; (**e**) Distribution of the offset distance between top-50 time series with a similar trend to the two methods; (**f**) Distribution of the offset distance between top-100 time series with a similar trend from to two methods.

**Table 3.** Street-views of TAZs retrieved by DTW and the proposed method.

**Table 3.** *Cont.*

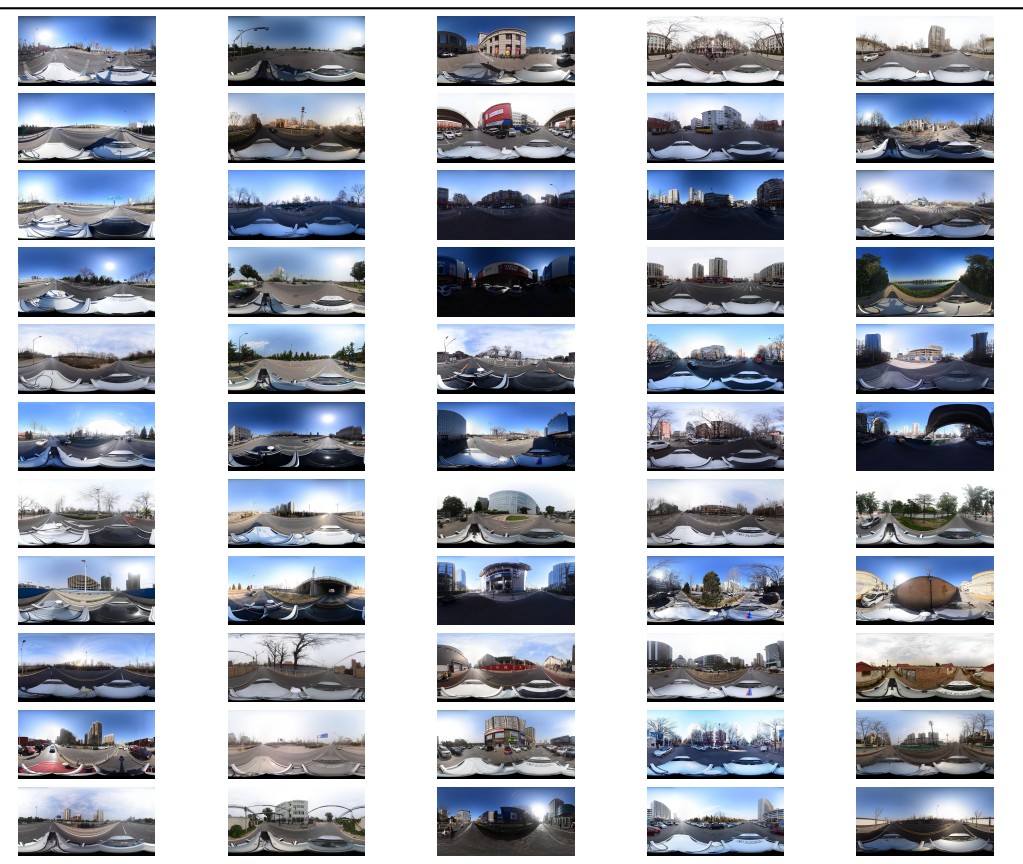

## 5. Discussion

### 5.1. Trade-Off between Information Loss and Search Efficiency

The essence of similar multi-dimensional time series searches is to consider features embedded in all data dimensions holistically. In this regard, it has to trade-off between information loss and search efficiency in that the efficiency of indexing decreases as the number of data dimensions increases [69]. In our proposed method, spatial objects can keep almost all information of the raw time series data since the number of spatial dimensions is identical to the number of dimensions of the raw time series data (at least in two- or three-dimensions). Consequently, further work will be conducted to verify how the indexing efficiency will decrease as spatial dimensions increase. Indeed, this problem is waiting to be addressed in almost all multi-dimensional time series mining methods that are under development.

### 5.2. Extension of Build-In Functions in Spatial Database

We verified that the result of the proposed method is highly similar to the query sequence, suggesting that the proposed method is reliable for time series similarity search. Nevertheless, the proposed method of similarity search in spatial databases does not meet the lower-bounding distance lemma, which means that it is an approximate similarity search method. Therefore, there are differences between the result sets obtained by the proposed method and the reference method in certain cases. Moreover, it is noteworthy that the difference is also partially because we used the Fréchet distance as a proxy of the DTW distance because the function for calculating DTW distance is not readily available in the PostGIS-enabled spatial database.

This limitation suggests that the proposed method is extensible in multiple ways. First, the built-in similarity measures in spatial databases can be replaced by other optimized algorithms. It is promising to construct and integrate other efficient distance functions

(such as optimized DTW and LCSS) into PostgreSQL with the PostGIS extension. Even though the DTW-based method was used as a reference in this study, it is not a ground-truth distance measure since every measure has its definition of "distance" or "(dis)similarity". In this sense, we should further consider a distance measure as proper or not depending on the applications of the time series similarity search. Second, spatial pruning is flexible to add more spatial features which are closely associated with time series features. The centroid and the MBR are equivalent to a very simple kind of upper and lower bounds of time series, i.e., *LB_Yi*. Future work will seek to design more advanced spatial features as a proxy of tighter lower bounds (such as *LB_Keogh*) for similar time series searches. Moreover, since the spatial database is only readily applicable to two-dimensional and three-dimensional time series, a major future challenge is that the spatial characteristics associated with high-dimensional time series are not as intuitive as that in 2/3-dimensional time series. It might be nontrivial to design appropriate spatial indicators for the pruning of high-dimensional time series to improve the generalizability of the proposed method.

Promisingly, with the rapid development of big data technology in recent years, many scholars have been applying big data frameworks such as Spark to establish distributed time index and to achieve parallel similar series retrieval [22,75]. Along with the further development of similarity measures and indexing methods, we believe that the comprehensive integration of the big data framework with the spatial databases will also play an important role in the field of similar time series searches.

### 5.3. Scalability of the Proposed Method

In practice, the scalability of methods for similar time series searches is essential. Despite that our proposed method primarily address the high dimensionality of large volume time series data, we also evaluate its performance upon time series data with different sequence length (see Table 4) as well as a different number of time series (see Table 5). By randomly sampling a subset of time series using predefined scale factors on sequence length and number, respectively, we generate datasets with different lengths and numbers of time series. Our result confirmed that the efficiency improvement, $\eta$, is substantial for a dataset consisting of shorter time series, while the consensus, $\rho$, is kept relatively high. It implies that as the length of the time series increases, the filtering ability of our proposed method based on the spatial features of time series data is less significant. Regarding the size of the time series dataset, our proposed method indeed demonstrated good scalability, as the efficiency improvement, $\eta$, is much higher on a dataset with a large number of time series. More rigorous evaluation might be done to provide solid evidence on the scalability of the proposed method.

**Table 4.** Scalability on datasets with different lengths of time series.

| Name | Scale Factor | Strict Similarity | | Trend Similarity | |
|---|---|---|---|---|---|
| | | $\rho$ | $\eta$ | $\rho$ | $\eta$ |
| CharacterTrajectories | 0.1 | 0.526 | 10.642 | 0.635 | 1.388 |
| | 0.5 | 0.409 | 2.711 | 0.788 | 0.466 |
| | 1.0 | 0.392 | 1.559 | 0.673 | 0.246 |
| EthanolConcentration | 0.1 | 0.105 | 5.433 | 0.500 | 2.247 |
| | 0.5 | 0.123 | 2.586 | 0.500 | 0.585 |
| | 1.0 | 0.116 | 1.494 | 0.500 | 0.368 |

**Table 5.** Scalability on datasets with different numbers of time series.

| Name | Scale Factor | Strict Similarity | | Trend Similarity | |
|---|---|---|---|---|---|
| | | $\rho$ | $\eta$ | $\rho$ | $\eta$ |
| CharacterTrajectories | 0.1 | 0.477 | 0.284 | 0.750 | 0.085 |
| | 0.5 | 0.478 | 0.869 | 0.673 | 0.175 |
| | 1.0 | 0.392 | 1.559 | 0.673 | 0.246 |
| EthanolConcentration | 0.1 | 0.273 | 0.286 | 0.615 | 0.090 |
| | 0.5 | 0.154 | 0.911 | 0.500 | 0.245 |
| | 1.0 | 0.116 | 1.494 | 0.500 | 0.368 |

*5.4. Similarity Search of Spatial Trajectories*

Our proposed method measures the similarity between spatial trajectories in the spatial database as a proxy of the similarity between raw time series. Indeed, with the rapid accumulation of big geospatial data, a lot of studies have already investigated the similarity search of spatial trajectories [76] in situations where databases are not required. Compared with time series data (and its converted spatial objects), spatial trajectory data contains richer information such as timestamp, moving speed, and direction for defining similarity measures. However, such information is often discarded to find trajectories with similar spatial characteristics. In this regard, the similarity between spatial trajectories is usually measured by either the warp-based distance (including DTW, LCSS, and EDR distances) [77,78], or/and the shape-based distance (including Hausdorff and Fréchet distances) [79], which are consistent with our proposed method. Additional distance measures for trajectory data include the integral of the distance from points on one trajectory to another trajectory (such as the one-way distance, and the symmetrized segment-path distance [80]), and the overlay of two trajectories in the 2D plane (such as the locality in-between polylines distance [81]), which might be also applicable to our analytical framework. It is also noteworthy that integrating distance in the temporal dimension is essential for similar trajectory searches if the sampling frequency varies in trajectory data. In such a case, the spatio-temporal similarity score must be applied [82].

A similar trajectory search is often implemented as a dynamic programming process, with a computational complexity of $O(mn)$. Similar to time series search, the pruning and local filtering strategies are often applied to boost the speed of similar trajectory search [79]. Moreover, specific storage and indexing methods for trajectory data have also been proposed, such as the segment-based partition and indexing [83], and the pivot-based metric indexing [84]. Recently, a few scholars have also applied deep learning methods for trajectory representation and computation. For example, ref. [85] implemented a trajectory vector extraction method based on RNN, which converted the trajectory distance into a vector distance calculation. Ref. [86] introduced an attention mechanism in the GRU sequence auto-encoder model to represent trajectories containing temporal features, and used the vector representation of each trajectory to calculate trajectory similarity scores. These newly developed techniques could shed new light on the storage and similarity search of time series data in both non-spatial and spatial databases.

## 6. Conclusions

This study introduced a novel similarity search method that leverages the power of spatial databases to support similar multi-dimensional time series searches. Multi-dimensional time series are represented as spatial objects and stored in spatial databases. Similar time series can be quickly retrieved with the assistance of spatial indexes built into spatial databases. Empirical experiments also confirmed that spatial databases can quickly prune time series using their spatial indexes and built-in functions, especially in cases where the retrieval number is small.

The main contribution of this study is to bring a novel spatial perspective to deal with the reliability, efficiency, and ease-of-use challenges in the similarity search of multi-dimensional time series. Compared with the commonly used GEMINI-based and lower-

bounding-based indexing methods, searching similar time series within the spatial database has several advantages. First, a database-based similarity search can ensure that the dimensionality of spatial objects is consistent with the raw time series before the conversion. Second, the indexing process of database-based search needs neither feature transformation nor upper/lower-bound calculation and therefore can support the efficient similarity search for multi-dimensional time series. Once the time series has been represented as spatial objects and stored in the spatial database, the spatial calculation functions integrated with the database can be directly used to calculate the spatial features by SQL statements. In doing so, it becomes simple and convenient to conduct a similarity search for users from the domain of geography who are familiar with spatial databases.

**Author Contributions:** Chaogui Kang designed the study. Zheren Liu implemented the proposed method and conducted the data analysis. Zheren Liu, Chaogui Kang, and Xiaoyue Xing analyzed the results; Chaogui Kang, Xiaoyue Xing, and Zheren Liu. wrote the main manuscript text. All authors reviewed the manuscript. All authors have read and agreed to the published version of the manuscript.

**Funding:** This work was partially supported by the National Natural Science Foundation of China (Grant No. 41601484 and 41830645), the National Key Research and Development Program of China (Grant No. 2017YFB0503604 and 2019YFE0106500), and the Open Fund of Hubei Luojia Laboratory (Grant No. 220100056).

**Data Availability Statement:** The authors declare that all data and code supporting the findings of this study are adequately disclosed within the article. Specifically, the four datasets from the UCR Time Series Classification Archive are available at https://www.cs.ucr.edu/%7Eeamonn/time_series_data_2018/ (accessed on 14 April 2022). The "BeijingDynamics" dataset can be shared upon reasonable request. The code used to reproduce the results are available at https://figshare.com/s/f2f74937ed2a4f6a5ecf (accessed on 22 October 2022).

**Conflicts of Interest:** The authors have no competing interests to declare that are relevant to the content of this article.

## Abbreviations

The following abbreviations are used in this manuscript:

| | |
|---|---|
| DTW | Dynamic Time Warping |
| LCSS | Longest Common Subsequence |
| GEMINI | Generic Multimedia Indexing |
| MBR | Minimum Bounding Rectangle |
| TAZ | Traffic Analysis Zone |

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
