# Peer review of "Querying Similar Multi-Dimensional Time Series with a Spatial Database"

_ijgi, doi:10.3390/ijgi12040179_

Round 1

Reviewer 1 Report

 This paper presents a new method for querying the similarity between multi-dimensional time series in databases. The authors conducted case studies using real data to test their methods. Although the manuscript has the potential to become a useful reference in the field, I have a few suggestions:

1. My understanding is that the authors treat multi-dimensional time series similar to trajectories in mobility analysis. Therefore, the literature review should be expanded to include previous work on trajectory similarity measures instead of only focusing on regular time series.

2. The authors explained that one of the motivations for their research is that DTW and other measures have a high time complexity. However, based on Figure 7, the performance of their method was not satisfactory. I am not entirely convinced by the results, so I suggest that the authors be more specific when explaining the scientific merits of their work.

3. The English was fine, but certain sentences can be lengthy and overly complicated. The following sentence is an example:

In order to demonstrate the fidelity of the proposed method, we justified the correlation between temporal features of the raw time series and spatial features of the corresponding spatial objects both theoretically and empirically

4. Please explicitly discuss the spatial and temporal efficacy of Frechet and other reference methods using the big O notation.

Overall, the manuscript has potential but could benefit from some revisions.

Author Response

  1. My understanding is that the authors treat multi-dimensional time series similar to trajectories in mobility analysis. Therefore, the literature review should be expanded to include previous work on trajectory similarity measures instead of only focusing on regular time series.

Thanks for the suggestion. We agreed that trajectory similarity measures could provide much insight into the calculation of shape differences between time series in the spatial database. Therefore, we have added subsection 5.4 in the "Discussion" section to introduce previous work on trajectory similarity measures and their connections with our proposed method.

  1. The authors explained that one of the motivations for their research is that DTW and other measures have a high time complexity. However, based on Figure 7, the performance of their method was not satisfactory. I am not entirely convinced by the results, so I suggest that the authors be more specific when explaining the scientific merits of their work.

Thanks for raising this question. We apologize that we didn't highlight our point clearly in Figure 7. The performance of our proposed method is lower than the DTW-based method for a large k. However, we have found that "the computational complexity of Fréchet distance and that of DTW distance are theoretically the same" (lines 468-470), which are both O(mn). However, in our study "LB_Keogh is used as a constraint for accelerating the C program implemented DTW-based search method, while the Fréchet distance function in PostgreSQL is not optimized" (lines 470-472). Due to this fact, our proposed method is slower than the DTW-based method. By replacing the Fréchet distance function with the optimized DTW distance, our proposed method will yield better performance than the DTW-based method (see the orange line in Fig 7a and 7d, and lines 481-483: "The combined method that uses spatial features for pruning and generating set S_2k and calculates exact LB_Keogh constrained DTW distance to obtain the result set S_k always yields lower time cost than other methods").

  1. The English was fine, but certain sentences can be lengthy and overly complicated. The following sentence is an example:

“In order to demonstrate the fidelity of the proposed method, we justified the correlation between temporal features of the raw time series and spatial features of the corresponding spatial objects both theoretically and empirically”

Thanks for the suggestion. We have revised the sentence as "In order to demonstrate the validity of the proposed method, we analyzed the correlation between temporal features of the raw time series and spatial features of their corresponding spatial objects theoretically and empirically".

  1. Please explicitly discuss the spatial and temporal efficacy of Frechet and other reference methods using the big O notation.

Thanks for the suggestion. We have explicitly discussed the temporal complexity of DTW distance and Frechet distance using the big O notation in Section 3.1.2: "In terms of computational complexity, the DTW distance and the Fréchet distance are the same to each other, which is O(mn)" (lines 301-303).

Reviewer 2 Report

In this paper, the authors proposed an approach to search for similar multi-dimensional time series with spatial databases, which can store them as spatial objects in a spatial database and search for similar time series based on their spatial features. I think it is exciting and suggest the authors revise some sentences. In sum, I argue that the paper can be accepted after modification. 

Author Response

I think it is exciting and suggest the authors revise some sentences. 

Thanks for your suggestion. We have done an additional editorial revision.

Reviewer 3 Report

A good manuscript. I hope to see more details and evaluations about Table 3.

I need to put forward that it has lost some essential, state-of-the-art, and closely related literature contributions as far as I know. Please compare and discuss. For example, TSSEARCH: Time Series Subsequence Search Library released in 2022 is an open-source code bank for subsequence similarity search. Grounded also on DTW, it provides some advanced improvements. Another approach to measuring similarity in time series is also published in 2022: based on temporal, statistical, and spectral domain feature extraction (like you did), the use of the self-similarity matrix and novelty function can perform black-box similarity analysis of single-dimensional or multidimensional time series (Feature-Based Information Retrieval of Multimodal Biosignals with a Self-Similarity Matrix). In this publication, the method was not only applied to physiological signals but also verified on datasets in various fields such as finance, geo-sensing, and sociology.

I have no disagreement with the content. Regarding the English expression, I still found some small grammar flaws and Typos, please check again.

Author Response

1. I hope to see more details and evaluations about Table 3.

Thanks for the suggestion. We have elaborated on how the representative street-view image was selected for each TAZ in this Table, "Note that we collect the street-view images of every 200m×200m grid within a TAZ, and select the scene with the most frequent appearance to represent its typical physical characteristics" (lines 517-519). In this table, we have two major findings: (1) Several TAZs are the same for both the DTW-based method and our proposed method (see the first few lines of images); (2) The representative street-view images for TAZs that are different for the DTW-based method and our proposed method (see the last few lines of images) are visually similar.

2. I need to put forward that it has lost some essential, state-of-the-art, and closely related literature contributions as far as I know. Please compare and discuss. For example, TSSEARCH: Time Series Subsequence Search Library released in 2022 is an open-source code bank for subsequence similarity search. Grounded also on DTW, it provides some advanced improvements. Another approach to measuring similarity in time series is also published in 2022: based on temporal, statistical, and spectral domain feature extraction (like you did), the use of the self-similarity matrix and novelty function can perform black-box similarity analysis of single-dimensional or multidimensional time series (Feature-Based Information Retrieval of Multimodal Biosignals with a Self-Similarity Matrix). In this publication, the method was not only applied to physiological signals but also verified on datasets in various fields such as finance, geo-sensing, and sociology.

Thanks for the suggestion. We have extended our review of related works in Sections 1 and 2.1. Above references have been discussed where they are relevant.

3. Regarding the English expression, I still found some small grammar flaws and Typos, please check again.

Thanks for the suggestion. We have double-checked and revised the manuscript.